# Back-Projected Signal-Based Self-Interferometric Phase Analysis Technique for Sea Surface Observation Using a Single Scatterometer System

**DOI:** 10.3390/s23063049

**Published:** 2023-03-12

**Authors:** Ji-hwan Hwang, Duk-jin Kim

**Affiliations:** 1Research Institute of Basic Sciences, Seoul National University, Seoul 88026, Republic of Korea; 2School of Earth and Environmental Science, Seoul National University, Seoul 88026, Republic of Korea

**Keywords:** self-interferometric phase, back-projection, wind speed, scatterometer

## Abstract

This manuscript presents a self-interferometric phase analysis technique for sea surface observation using a single scatterometer system. The self-interferometric phase is proposed to complement the imprecise analysis results due to the very meager signal strength measured at a high incident angle of more than 30°, which is a vulnerability of the existing analysis method using the Doppler frequency based on the backscattered signal strength. Moreover, compared to conventional interferometry, it is characterized by the phase-based analysis using consecutive signals from a single scatterometer system without any auxiliary system or channel. To apply the interferometric signal process on the moving sea surface observation, it is necessary to secure a reference target; however, this is hard to solve in practice. Hence, we adopted the back-projection algorithm to project the radar signals onto a fixed reference position above the sea surface, where the theoretical model for extracting the self-interferometric phase was derived from the radar-received signal model applying the back-projection algorithm. The observation performance of the proposed method was verified using the raw data collected at the Ieodo Ocean Research Station in Republic of Korea. In the observation result for wind velocity at the high incident angles of 40° and 50°, the self-interferometric phase analysis technique shows a better performance of a correlation coefficient of more than about 0.779 and an RMSE (root-mean-square error) of about 1.69 m/s compared to the existing method of a correlation coefficient of less than 0.62 and RMSE of more than 2.46 m/s.

## 1. Introduction

A scatterometer system has operated to measure and analyze the radar backscatter from the earth’s surface and has developed into the spaceborne or airborne systems and the ground-based system [1,2,3,4]. The satellite systems, from SEASAT as the first satellite for ocean observation to the latest systems, such as the QuikSCAT- and RapidSCAT-mounted SeaWinds instrument, have steadily operated to measure the physical oceanographic parameters, such as ocean wind [5,6,7]. Whereas the ground-based system has been used as an auxiliary means to conduct a restricted mission, such as in situ data acquisition for calibration and validation of the satellite or airborne system [8]. However, the ground-based system that has developed into customized systems through various field campaigns has high potential in terms of utilization. It is, namely, possible to implement a cost-effective system for specialized observation in a specific area and even measure various radar parameters, such as radar backscatters, distance (or depth), and Doppler frequency simultaneously [9,10,11].

In a previous study [11], to apply this potential to sea surface observation, we implemented a multifunctional scatterometer system based on the self-manufactured FMCW (frequency-modulated continuous wave) radar transceiver and validated its performance. It can simultaneously measure several radar parameters, such as radar backscatter (*σ°*), target distance (*R*), and Doppler frequency (*f_D_*), and analyze the changes in wave height or ocean wind using them. The ocean wind could be particularly estimated from the Doppler frequency (*f_D_*) collected at the oblique incident angle of 10° to 50°, where the Doppler frequency was extracted by the automatic peak search based on the radar received signal strength. However, because the backscattered signal strength from the sea surface rapidly decreases as the incident angle increases, there was a vulnerability that the accuracy of the extracted Doppler frequency and the estimated wind velocity fall together. In addition, the radio wave of scatterometer is resonant with the wind-generated ocean wave, which has a wavelength of a few cm or mm, and the Bragg scattering occurs in this process [12,13]. For sea surface observation, the Bragg scattering is usually measured at relatively high incident angles of more than 30° and has a characteristic highly correlated to the ocean wind. Therefore, the data measured at relatively high incident angles are even more needed to precisely retrieve the ocean wind from the radar-backscattered signals [6,7,14].

In this study, we propose a phase-based analysis technique, which is robust even at the high incident angle, to complement this point. The proposed method analyzes the phase difference—the so-called ‘self-interferometric phase’—between consecutively received signals of a single channel, unlike the conventional interferometric analysis method using multiple channels, to observe the sea surface movement. Tiny changes in the target distance (*R*) due to the sea surface movement can be extracted through phase analysis of the radar-received signals measured at a regular time interval (Δ*t*). However, to apply the interferometric phase for the sea surface monitoring, a fixed target for phase analysis should be first designated. Unfortunately, it is nearly impossible on the constantly moving sea surface. Therefore, we apply the back-projection algorithm to help the phase analysis of the radar-received signals. The back-projection algorithm is generally used for SAR (synthetic aperture radar) image focusing, whereas, in this study, it is for providing a reference grid to compare the phase components of the consecutive signals under the same condition. Applying the back-projection algorithm, all consecutively received signals can be projected onto the same reference grid with nothing to do with the sea surface movement; then we can stably compare their phase terms.

The proposed phase-based analysis technique for sea surface observation, particularly ocean wind retrieval, will be sequentially described in detail in this paper. Section 2 briefly shows the self-manufactured scatterometer system and its vulnerability. In Section 3, the self-interferometric phase model is theoretically derived by the radar signal model and back-projection algorithm [15,16,17,18]. Then, it is converted to a relative velocity for ocean wind retrieval. Moreover, in Section 4, this proposed technique is verified using the raw data measured by our scatterometer system and the in situ data collected at the Ieodo Ocean Research Station [19,20]. In particular, spectral characteristics of the self-interferometric phase data and correlation to the in situ data are analyzed to retrieve ocean wind speed or wave height. Finally, Section 5 shows the improved performance of the proposed method through the observation results applying the spectral characteristics and correlation function analyzed in the previous section.

## 2. Features of Sea Surface Observation Data (by the Existing Method)

This section presents the property of the ocean observation data collected by the self-manufactured multifunctional scatterometer in the previous study [11], as shown in Figure 1 and Table 1. This section briefly shows the vulnerability of the existing analysis method based on the radar-received signal strength through the radiometric responses and linear regression analysis for each incident angle, where the observation data were measured under the sea surface condition of less than WMO (world meteorological organization) sea state code 3 (i.e., it is less than 1.25 m of wave height and the capillary wave dominant) was used [21].

### 2.1. Radiometric Responses from Sea Surface

Figure 2 summarizes the angular responses and operating range of the existing and proposed methods for incident angles. The raw data measured by our scatterometer system were calibrated into the backscattering coefficients (*σ°_vv_*) with vertical polarization, as shown in Figure 2a, and show the angular response of the radar backscatter from the sea surface as well. The strength of the radar backscattered signals rapidly decreases to the incident angle of about 30° and then slowly down. Here, because the responses after 30° become very close to the system noise floor, it is hard to distinguish sea clutter and noise. Unfortunately, to observe physical oceanographic parameters such as sea wind speed, it is necessary to record the observation data, including the Bragg resonance scattering, which reflects radar backscattering for the wind-generated short capillary waves, and these are mainly measured at the high incident angles of more than 30°. The dependency of radar backscatter versus incident angle for moderate wind speed is also ascertainable in [12,22,23,24]. Moreover, Figure 2b is conceptually drawn from the operating range of our multi-functional scatterometer system. The existing method based on the backscattered signal strength can be applied to the data measured at all incident angles; however, only the data within 0° to 30° is available for ocean observation. This vulnerability is complemented by the phase-based method using a self-interferometric phase. The proposed technique can be applied to all incident angles; in particular, it will be used to retrieve the sea wind speed.

### 2.2. Correlation of Doppler Frequency to Wind Speed

Doppler frequency (*f_D_*), one of the radar measurement parameters, is an effective indicator for estimating wind speed, but it is extracted by the existing method based on signal strength. Therefore, the Doppler frequency extracted from the raw data by automatic peak search based on the signal strength is critically affected by the low signal strength at high incident angles.

Figure 3 shows the effects due to the change in the signal strength for each incident angle through the linear regression analysis. Figure 3a–e are the linear regression analysis results using the wind speed (*w_s_* (*f_D_*)) estimated by the Doppler frequency and the wind speed (*w_s_* (in situ)) observed at the Ieodo Ocean Research Station, and Figure 2f shows the change in the correlation coefficients for each incident angle. The critical change is clearly shown at the high incident angle of more than 30°. It becomes the highest correlation coefficient at the incident angle of 30° and then rapidly decreases, where the correlation coefficients at the incident angles of 40° and 50° are about 0.619 and 0.287, respectively. (Data process using the regression analysis will be presented further in the following sections.) The low signal strength at the high incident angles of 40° and 50° critically affects the Doppler frequency-extracting process and, as a result, the correlation between Doppler frequency and wind speed also rapidly decrease. Therefore, we propose the phase-based analysis method using the self-interferometric phase as an additional radar parameter to complement this weak point. The theoretical signal model for extracting and analyzing the self-interferometric phase is derived in the next section.

## 3. Signal Model for Self-Interferometric Phase

Of the observation methods able to analyze changes in the sea surface, the observation method using Doppler frequency can effectively be applied to estimate the ocean wind corresponding to the radiometric responses within a specific range. However, because of the low signal strength at the high incident angles, as mentioned before, there is a restriction in usage. By contrast, the phase-based method using the self-interferometric phase can directly measure the delicate changes in the target distance (*R*) during a very short period (Δ*t*). This is presented further using the radar geometry for ocean observation, as shown in Figure 4. To establish the signal model for analyzing the self-interferometric phase, we first consider two cases of geometries with a short time interval (Δ*t*) as shown in Figure 3a,b. It shows the cases in which a fixed radar measures an arbitrarily moving target on the sea surface at an initial time (*t*_0_) and the next time (*t*_0_ + Δ*t*) after Δ*t*, respectively. Here, *R*_0_ is the reference target distance designated to back-project the radar-received signal and set considering the height (*h*) between the antenna and the mean sea level (e.g., *z* = 0). Moreover, Figure 3c shows the change in the target’s phase centers (*P_s_*_(1)_
*or P_s_*_(2)_) with the reference phase center (*P_g_*). The delicate change in the distances (*R*_(1)_
*or R*_(2)_) between the antenna and the target phase center is detectable within a range resolution (*δR ≈* [*R*_0_ − *δR*/2, *R*_0_ + *δR*/2]) because the phase component represents only –π to π. The self-interferometric phase model is theoretically derived using these radar geometries and the back-projected signal model as follows.

### 3.1. Back-Projected Signal Model onto the Sea Surface

The interferometric phase means the phase difference between two signals, which are measured at different positions or times. Our setup for ocean observation corresponds to the latter (refer to Figure 4) because the fixed radar system measures the moving sea surface in a very short period. Moreover, the radar backscattered signals measured by this setup should be back-projected onto the reference position on the sea surface to compare with the phase component of each signal. To derive the self-interferometric phase model, the FMCW (frequency-modulated continuous wave) signal model and the back-projected signal model are used as shown in Equations (1) and (2), where *f*_0_ is an operating frequency, *K_r_* is a chirp rate, *τ* is the target’s delay time (e.g., *τ* = 2*R*/*c*), *k*_0_ is a wave number, and α is the received signal strength or magnitude, respectively. Equation (1) shows the FMCW signal model as the frequency down-converted intermediate frequency and the signal model approximated by the propagated range, respectively [15]. Equation (2) describes the back-projected signal onto the reference position, which applies a computing process of the back-projection algorithm [17,18]. Here, the back-projection algorithm is used for extracting the phase components of the reference position rather than for imaging the radar signals.
(1)$$\begin{array}{l} s_{IF,r}\left(t\right)=\alpha \cdot \exp\left\{-j\left(2\pi f_{0}\tau +2\pi K_{r}t\tau -\pi K_{r}\tau^{2}\right)\right\}\\ \;\;\;≅\alpha \cdot \exp\left\{-j2k_{0}R\right\} \end{array}$$
(2)$$\begin{array}{l} f\left(x,y\right)=\int s_{IF,r}\left(t\right)\cdot h^{*}\left(t\right)\;dt\\ \;\;\;≅\alpha \cdot \exp\left\{-j2k_{0}\cdot \left(R-R_{0}\right)\right\}=\alpha \cdot \exp\left\{-j2k_{0}\cdot \Delta R\right\} \end{array}$$

The signal processing to get the back-projected signal is integrating the multiplication of the received radar signal of Equation (1) and the matched filter of *h*(*t*), where the reference distance (*R_0_*) is a set value (or known value) while the actual distance (*R*) is an unknown value. Therefore, we define that each point on the radar geometry is as Equation (3), e.g., *P_g_*(*x*,*y*,0) is the reference point to back-project the received radar signal onto the sea surface, *P_s_*_,(1)_(*x*_1_,*y*_1_,*z*_1_) is the actual target position at an initial time (*t*_0_), *P_s_*_,(2)_(*x*_2_,*y*_2_,*z*_2_) is the shifted target position at the next time step (*t*_0_ + Δ*t*), and *P_A_*(0,0,*h*) is the antenna position with a height (*h*) from the mean sea level (*z* = 0). Moreover, the actual target positions (*P_s_*_(1)_, *P_s_*_(2)_) can be represented with the reference position *P_g_*(*x*,*y*,0) and the residual components (*δx*_1_,*δy*_1_,*δz*_1_) or (*δx*_2_,*δy*_2_,*δz*_2_).
(3)$$\begin{array}{l} P_{g}\left(x,y,0\right)\\ P_{s,\left(1\right)}\left(x_{1},y_{1},z_{1}\right)=P_{s,\left(1\right)}\left(x+\delta x_{1},y+\delta y_{1},z+\delta z_{1}\right)\\ P_{s,\left(2\right)}\left(x_{2},y_{2},z_{2}\right)=P_{s,\left(2\right)}\left(\left(x+\delta x_{1}\right)+\delta x_{2},\left(y+\delta y_{1}\right)+\delta y_{2},\left(z+\delta z_{1}\right)+\delta z_{2}\right) \end{array}$$

The distance between the target and the antenna can be calculated using the defined points in Equation (3). Equations (4) and (5) show the reference range (*R*_0_) and the actual range (*R*_(1)_) at the initial time (*t*_0_), respectively. The range (*R*_(2)_) at the nest time (*t*_0_ + Δ*t*) after Δ*t* can be also derived in the same way.
(4)$$R_{0}\left(P_{g};P_{A}\right)=\sqrt{x^{2}+y^{2}+{\left(z-h\right)}^{2}}$$
(5)$$R\left(P_{s,\left(1\right)};P_{A}\right)=\sqrt{{\left(x+\delta x_{1}\right)}^{2}+{\left(y+\delta y_{1}\right)}^{2}+{\left(z+\delta z_{1}-h\right)}^{2}}$$

The back-projected signal onto the reference point is processed by the matched filtering on the back-projection algorithm applying the reference range (*R*_0_) and actual range (*R*) in Equations (4) and (5). Then, to analyze the phase term in the back-projected signal, the actual range (*R*) can be approximated by Taylor’s series expansion (e.g., *sqrt*(*m*^2^ + *n*) *≈ m* + *n*/2*m*), as shown in Equation (6). Moreover, the residual range (Δ*R*) of the back-projected signal can be derived by subtracting the reference range from the actual range, as shown in Equation (7).
(6)$$\begin{array}{l} R_{\left(1\right)}=\sqrt{\left(x^{2}+y^{2}+h^{2}\right)+\left\{2x\delta x_{1}+2y\delta y_{1}-2h\delta z_{1}+\delta x_{1}^{2}+\delta y_{1}^{2}+\delta z_{1}^{2}\right\}}\\ \;\;\;\;≅R_{0}+\frac{1}{2R_{0}}\left\{2x\delta x_{1}+2y\delta y_{1}-2h\delta z_{1}+\delta x_{1}^{2}+\delta y_{1}^{2}+\delta z_{1}^{2}\right\} \end{array}$$
(7)$$\Delta R_{\left(1\right)}=\left\{R_{\left(1\right)}-R_{0}\right\}=\frac{1}{2R_{0}}\left\{2x\delta x_{1}+2y\delta y_{1}-2h\delta z_{1}+\delta x_{1}^{2}+\delta y_{1}^{2}+\delta z_{1}^{2}\right\}$$

The residual range of the back-projected signal at an initial time (*t*_0_) is as shown in Equation (7). It is just reflected in the phase term (*Φ* = −2*k*_0_*·*Δ*R*) of the back-projected signal.

### 3.2. Self-Interferometric Phase Model

The radar-received received signals are transformed into the back-projected signals onto the set grid points as the reference sea surface, and the self-interferometric phase model can be derived from this back-projected signal. It can, namely, be analyzed from a phase difference between the residual phase of the back-projected signals [25,26].

Equations (8) and (9) show the actual range (*R*_(2)_) at the next time step after the intervals of Δ*t* and its residual range (Δ*R*_(2)_), respectively. Here, the residual components (*δx*,*δy*,*δz*) represent the summation of each residual component at the initial or next-time step, i.e., (*δx*,*δy*,*δz*) = (*δx*_1_ + *δx*_2_, *δy*_1_ + *δy*_2_,*δz*_1_ + *δz*_2_).
(8)$$R_{\left(2\right)}≅R_{0}+\frac{1}{2R_{0}}\left\{2x\delta x+2y\delta y-2h\delta z+\delta x^{2}+\delta y^{2}+\delta z^{2}\right\}$$
(9)$$\Delta R_{\left(2\right)}=\left\{R_{\left(2\right)}-R_{0}\right\}≅\frac{1}{2R_{0}}\left\{2x\delta x+2y\delta y-2h\delta z+\delta x^{2}+\delta y^{2}+\delta z^{2}\right\}$$

Then, the self-interferometric phase can be derived from a difference (Δ*R*_(2)_−Δ*R*_(1)_) between the residual ranges in the back-projected signals, as shown in Equation (10). It can convert into the vector form to clarify the correlation between the sea surface movement (*δ*_2_) and the observational condition (*R*_0_), where each vector represents $\bar{R_{0}}=x\hat{x}+y\hat{y}-h\hat{z}$, $\bar{\delta_{1,2}}=\delta x_{1,2}\hat{x}+\delta y_{1,2}\hat{y}+\delta z_{1,2}\hat{z}$, $\bar{\delta_{2}}=\bar{v}\Delta t$, and $\bar{v}$ is the velocity vector of the moving target phase center. The first term in the final equation becomes a significant value that represents the sea surface movements projected onto the reference range (*R*_0_). On the other hand, the latter two terms are ignorable because of too small values.
(10)$$\begin{array}{l} \Delta R_{\left(2\right)}-\Delta R_{\left(1\right)}\\ ≅\frac{1}{2R_{0}}\left\{2\left(x\delta x_{2}+y\delta y_{2}-h\delta z_{2}\right)+2\left(\delta x_{1}\delta x_{2}+\delta y_{1}\delta y_{2}+\delta z_{1}\delta z_{2}\right)+\left(\delta x_{2}^{2}+\delta y_{2}^{2}+\delta z_{2}^{2}\right)\right\}\\ =\frac{1}{2R_{0}}\left\{2{\bar{R}}_{0}\cdot {\bar{\delta }}_{2}+2{\bar{\delta }}_{1}\cdot {\bar{\delta }}_{2}+{\left|{\bar{\delta }}_{2}\right|}^{2}\right\}=\left\{{\hat{R}}_{0}\cdot \left(\bar{v}\Delta t\right)+\frac{{\bar{\delta }}_{1}\cdot \left(\bar{v}\Delta t\right)}{R_{0}}+\frac{{\left|\bar{v}\Delta t\right|}^{2}}{2R_{0}}\right\} \end{array}$$

To extract the self-interferometric phase from the back-projected signals, it conducts a multiplication of the next signal and a complex conjugate of the initial signal, as shown in Equation (11). Then, we can analyze the physical oceanographic parameters related to the sea surface movements from the phase term of the processed signal in Equation (12). Besides that, Equation (12) shows the relation between the interferometric phase (ΔΦ) and the sea surface movement approximated from Equation (10), where *v_r_* represents the relative velocity of the moving target phase center projected on the observational angle.
(11)$$f_{\left(2\right)}\left(x,y\right)\cdot f_{\left(1\right)}^{*}\left(x,y\right)=\beta \cdot \exp\left\{-2jk_{0}\left(\Delta R_{\left(2\right)}-\Delta R_{\left(1\right)}\right)\right\}$$
(12)$$\Delta \Phi =-2k_{0}\left(\Delta R_{\left(2\right)}-\Delta R_{\left(1\right)}\right)≅-2k_{0}\left\{{\hat{R}}_{0}\cdot \left(\bar{v}\Delta t\right)\right\}=-2k_{0}\cdot \left(v_{r}\Delta t\right)$$

Equation (13) is an inverted formula to predict the sea surface movement from the self-interferometric phase (ΔΦ) bounded from –π to π. Thus, ambiguity may occur in the sea state exceeding this condition, only when the change (e.g., Δ*R*_(2)_ − Δ*R*_(1)_) is over more than half of the wavelength per the time interval (Δ*t*) due to the rough sea state. Moreover, because the time interval depends on the pulse repetition frequency (*f_PRF_*) of the scatterometer system (refer to Table 1), the maximum detectable range of the relative velocity (*v_r_*) can be easily computed through Equation (13).
(13)$$v_{r}≅-\frac{\lambda_{0}}{4\pi }\frac{\Delta \Phi }{\Delta t},\left(\Delta \Phi \in \left[-\pi ,\pi \right]\right)$$

In summary, this theoretical model for the self-interferometric phase was derived from the back-projected signal, which is projected to the fixed reference grid through the back-projection algorithm. Moreover, the relationship to the sea surface movement was converted to the relative velocity. In addition, because the detectable range of the relative velocity is determined by the short wavelength of X-band operating frequency, the observable range of the sea state may be restricted to the relatively smooth sea state, in which the capillary wave is mainly dominant.

## 4. Analysis of Self-Interferometric Phase Data

The self-interferometric phase model derived above is verified using the raw data measured by the self-manufactured FMCW-scatterometer system, which has a pulse repetition frequency (*f_PRF_*) of 100 Hz, i.e., time interval Δ*t* = 10 ms (refer to Table 1). This section first shows that the interferometric phase can be extracted from the back-projected signals as the reconstructed image on the reference grid. Then, it analyzes the property of the collected phase data through spectrum analysis as well as regression analysis. Here, the raw data collected by the field campaign conducted at the Ieodo Ocean Research Station is used for cross-validation.

### 4.1. Extracting the Self-Interferometric Phases

Figure 5 shows the process of extracting the self-interferometric phase from the back-projected signals. The received signals are projected onto the fixed grid above the sea surface, as shown in Figure 5a, then the phase terms of these back-projected signals are compared on the same grid. To effectively show each back-projected signal (e.g., *f*_(1)_(*x*,*y*) or *f*_(2)_(*x*,*y*) in Equation (11)), their magnitude and phase parts were separately drawn onto the same grid, as shown in Figure 5b,c. Moreover, Figure 5d shows the coherence and phase difference, which are calculated by the Pearson product–moment correlation coefficient [27], where the phase difference becomes just the self-interferometric phase (refer to Equations (11) and (12)). The phase data that will be used for sea surface observation can be collected near the radar beam center, as shown in Figure 5d. Besides that, for computing, the antenna’s height (e.g., *h* ≈ 26 m) can be known by the target distance (*R*) extracted from the raw data measured at the nadir angle.

### 4.2. Property of the Self-Interferometric Phase Data

Figure 6 shows the time-series self-interferometric phase data extracted from the raw data measured at the incident angles of 0° to 50°. In particular, to analyze the spectral feature of the extracted phase data, we tried to separate the low- and high-frequency components through the digital filter process.

For the low-pass filtered data (‘LPF’ in Figure 6), a filtering order of 50 and a normalized Nyquist cutoff frequency of 0.01 were applied to extract the temporal changes in the time-series phase data collected at different observation conditions. Moreover, it has a very similar trend regardless of the observation angle, as shown in Figure 6f. Therefore, we can infer that the ‘LPF’ data reflect the radiometric responses related to the horizontal component of the sea surface movements through the back-projection and digital filter processes.

For the high-pass filtered data (‘HPF’ in Figure 7), the band-pass filter property of a filtering order of 50 and a normalized Nyquist cutoff frequency of [0.1, 0.65] was applied to remove low-frequency components and abnormal data as an outlier simultaneously. Figure 7a–e shows the high-frequency component only of the above time-series phase data, where the high-pass filtered data have a dispersive feature depending on the sea state or the incident angles. In particular, these data are reprocessed through statistical analysis, where the standard deviation (*σ*) to statistically analyze the dispersiveness and the weight factor (1/*cosθ_i_*) to compensate for the effect due to the incident angle were applied, as shown in Figure 7f. Moreover, applying the weight factor gives additionally an effect that filters a vertical component of the sea surface movement. In Figure 7f, these data show a similar statistical trend related to changes in wave heights. However, to measure this, the existing method of collecting measurement data near the nadir angle shows better performance.

In summary, Figure 6 and Figure 7 clearly show the spectral feature of the time-series self-interferometric phase data separated by the vertical or horizontal component. Based on this property, we try to analyze the correlation between the filtered data and the physical oceanographic parameters. For example, the ‘LPF’ data may be used to analyze oceanographic parameters such as wind speed related to the horizontal component of the sea surface movement, and the dispersiveness feature of the ‘HPF’ data may be auxiliarily applied to observe the wave height.

### 4.3. Correlation to Physical Oceanographic Parameters: (e.g., Wind Speed, Significant Wave Height)

At first, using the ‘LPF’ data, the correlation to wind speed was analyzed, as shown in Figure 8. Figure 8a is the wind speed (*w_s_*) as in situ data measured at Ieodo Ocean Research Station, which is compensated for the directional component of wind vectors considering the radar look-angle on the East side. Here, the minus sign means an opposite direction component over the look-angle. Moreover, the ‘LPF’ data are converted into the relative velocity (*v_r_*) of the sea surface movement by Equation (13), as shown in Figure 8b. Then, the correlation of the relative velocity versus the wind was analyzed, as shown in Figure 8c. It shows that the relative velocity (*v_r_*) has a linearly high correlation to the wind speed (*w_s_*) and, through the least square method [28], confirmed that it has the property of a linear function such as *w_s_* = 26.5*·v_r_*.

Using the relative velocity (*v_r_*) converted from the self-interferometric phase (ΔΦ), the data measured at all observational angles were analyzed by the linear regression analysis, which is a widely used statistical method to analyze the correlation between the radar and oceanographic parameters [29,30]. Figure 9 shows the results of linear regression analysis between the insitu wind speed (*w_s_*(insitu)) and the estimated wind speed (*w_s_*(ΔΦ)). Excepting the nadir angle data in Figure 9a, Figure 9b–f show the linearly high correlation. Especially at the high incident angle of 40° to 50°, it shows correlation coefficients of more than about 0.79.

Next, using the ‘HPF’ data related to the vertical movement on the sea surface, the correlation to wave height was analyzed, as shown in Figure 10. To retrieve physical oceanographic parameters such as the significant wave height (*H_sig_*) related to the vertical change, we adopted the conventional process to statistically calculate the significant wave heights, whose statistical definition is *H_sig_* = 4*σ* [31]. Figure 10a is the in situ data of the significant wave height calculated by the conventional process using a standard deviation (*σ*) of the measured wave height, and Figure 10b is the result of applying the same process to the ‘HPF’ data, which was measured near the nadir angle for observing the vertical component. Then, the correlation of the standard deviation of the ‘HPF’ data (*σ*(ΔΦ)) versus the significant wave height (*H_sig_*) was analyzed, as shown in Figure 9c. The result shows that the phase data has a linearly moderate correlation to the significant wave height and, through the least square method, confirms that it has the property of a linear function such as *H_sig_* =1.2·*σ*(ΔΦ). Moreover, the linear regression analysis process was equally applied to the ‘HPF’ data, but the correlation diagrams were omitted; the result only as the correlation coefficient is presented in the next section.

Note that, in this proposed analysis method using the self-interferometric phase, ambiguity may occur under the rough sea state when the maximum vertical or horizontal change is over half of the wavelength (*λ*_0_/2 *≈* 1.5 cm) during the time interval of about 10 ms (refer to Equation (13)). It namely means that the moving sea surface has a velocity of 1.5 m/s, theoretically.

## 5. Observational Results

The above results confirmed that the time-series interferometric phase data has a high correlation to the physical oceanographic parameters, such as wind speed or significant wave height. Using the temporal change of a low-frequency component and the dispersiveness of a high-frequency component of the phase data, we can retrieve the wind speed and significant wave height from the relatively smooth sea surface condition, which has a significant wave height of less than 1 m.

To verify the enhanced performance at the high incident angles of more than 30°, the wind speed was estimated by the proposed and existing methods, respectively, where the Doppler frequency (*f_D_*) extracted by the automatic peak search based on the peak signal strength was used as the existing method. Moreover, the significant wave height was also inverted from the standard deviation of the ‘HPF’ data related to the vertical component of the sea surface movement. However, this result is used only as an auxiliary for analyzing the performance of the proposed technique. In this case, the existing method is more efficient because the distance (*R*) can directly measure the changes in the wave height at the nadir angle.

### 5.1. Wind Speed

Figure 11 shows the comparison results of wind speed retrieved by self-interferometric phase and Doppler frequency. Figure 11a,b are examples for comparing the enhanced performance at the high incident angles, where the data measured at the incident angles of 40° and 50° are used only. These show how effective the proposed method is in compensating for the shortcoming of the existing method. Figure 11c is the final observation result retrieved by the data measured at all incident angles. Moreover, Figure 11d shows the variation of correlation coefficients at each incident angle before and after applying the proposed method. Here, the relative velocity (*v_r_*) converted by Equation (13) and the correlation function (e.g., *w_s_* = 26.5*·v_r_*) were equally applied for the wind speed retrieval.

The wind speed retrieved by the proposed method agrees well, even at the high incident angle, with the in situ data and keeps a correlation coefficient of about 0.776 at all incident angles on average. For quantitative performance analysis, the root-mean-square error (RMSE) comparing the difference between the retrieved wind speed and in situ data is additionally used with the correlation coefficient because the correlation coefficient theoretically does not mean the accuracy of analysis results. The observation result by the proposed method shows an RMSE of about 1.73 m/s. In particular, at the high incident angles, the results of the proposed and existing methods are an RMSE of 1.69 m/s and 2.27 m/s, respectively.

### 5.2. Significant Wave Heights

The standard deviation (*σ*(ΔΦ)) of the interferometric phase data collected near the nadir angle (at the incident angles of 0° and 10°) was used to retrieve the significant wave height (*H_sig_*), where the standard deviation was calculated by applying the time window of 1 h length with 100 data samples, and the correlation function (e.g., *H_sig_* =1.2*·σ*(ΔΦ)) was used for retrieving the significant wave height.

Figure 12 shows the comparison results of the significant wave height (*H_sig_*) retrieved by the interferometric phase (ΔΦ) and target distance (*R*). The significant wave heights retrieved by the proposed method have an RMSE of 8.03 cm and a correlation coefficient of about 0.814, respectively. For this case, because the change in wave height is directly measured at the nadir angle by the distance (*R*), the existing method shows better performance of an RMSE of 7.03 cm and a correlation coefficient of 0.873. In other words, this is due to a difference between the estimates and the measurements, we confirmed that the estimates also show an analytical ability comparable to the measurements.

Finally, the observational performances for wind speed and significant wave height are summarized again in Table 2, and in particular, the performance improvement at the high incident angles can be confirmed.

## 6. Conclusions

The self-interferometric phase (ΔΦ)-based analysis technique for sea surface observation was proposed to improve the operational efficiency under the observational condition with a high incident angle of more than 30°, which is hard to distinguish from system noise floor due to very low strength of the radar backscatter signal. Through the observation results for wind speed, we confirmed the enhanced performance of the proposed phased-based analysis technique, which shows a correlation coefficient of 0.776 and an RMSE of 1.692 m/s at all observational conditions. Especially at the high incident angle of more than 30°, it indicates much better stable performance than the existing method using Doppler frequency (*f_D_*). Moreover, we confirmed that the self-interferometric phase data are not only extracted from the back-projected signals but also the back-projection algorithm can be effectively applied to analyze the phase variation of the moving sea surface. Moreover, the proposed method can more effectively filter the radar backscatter signals near the beam center from the spatially or electrically unwanted signals because the self-interferometric phase information is collected from the fixed reference position given by the back-projection algorithm. As a result, the observation data stability or analysis performance could be improved.

Through various analysis results using actual raw data, this technique showed enough potential for sensing the delicate changes in the smooth sea surface, where the capillary wave is dominant. Moreover, the interferometric phase data, including all of the vertical and horizontal components of the sea surface movement, can simultaneously retrieve the physical oceanographic parameters, such as ocean wind and significant wave height, without any other processing or radar parameters. For significant wave height, the usage may be limited because the operating mode for altimeter is available. Nevertheless, in the retrieval of wind speed, it is very effective for analyzing the horizontal component even at the high incident angle because of using the signals projected onto the flat surface by the back-projection algorithm. Finally, it shows that this analysis technique using the self-interferometric phase data has not only enough performance to compensate for the vulnerability of sea surface observation at the high incident angles but also has a limitation of observational range due to a short wavelength of X-band frequency. Further study using a relatively lower frequency such as L- or C-bands may also be needed to extend the detectable range to more rough sea surfaces.

## Figures and Tables

**Figure 1 sensors-23-03049-f001:**
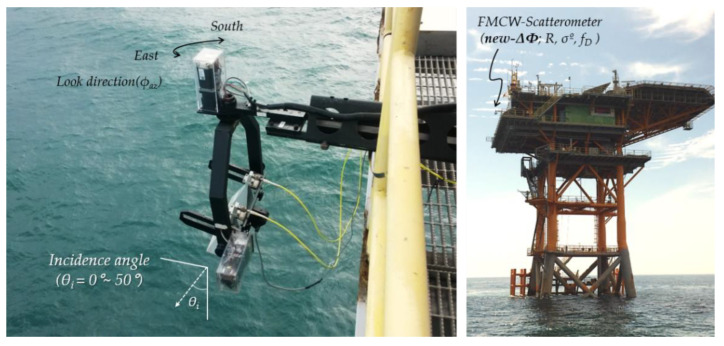
The appearance of the self-manufactured multifunctional FMCW-scatterometer system installed at the Ieodo Ocean Research Station located about 150 km from Jeju island of Republic of Korea.

**Figure 2 sensors-23-03049-f002:**
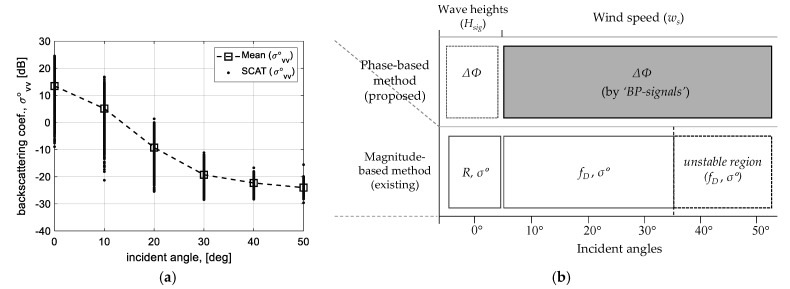
Radiometric angular responses from the sea surface and available range of radar measurement parameters: (**a**) backscattering coefficients (*σ°_vv_*) measured by self-manufactured scatterometer [11], and (**b**) comparison of the operating range of the proposed and existing method.

**Figure 3 sensors-23-03049-f003:**
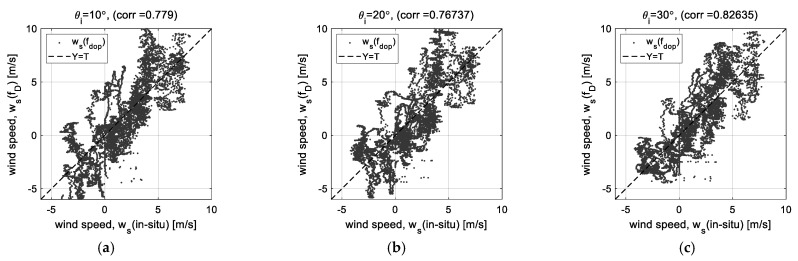
Results of the linear regression analysis of Doppler frequency (*f_D_*) to wind speed (*w_s_*): (**a**–**e**) correlation diagrams at incident angles of 10°~50°, and (**f**) variation of correlation coefficients for each incident angle.

**Figure 4 sensors-23-03049-f004:**
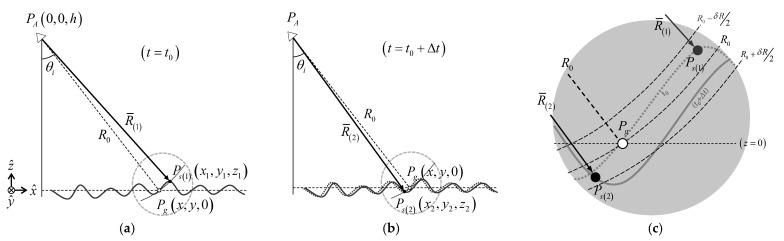
Radar geometries to establish the self-interferometric phase model: (**a**) an initial state at (*t* = *t*_0_), (**b**) the next state after Δ*t*, and (**c**) the ranges to target (*R*_(1)_, *R*_(2)_) compared with a reference range (*R*_0_) (where *δR* is a slant range resolution).

**Figure 5 sensors-23-03049-f005:**
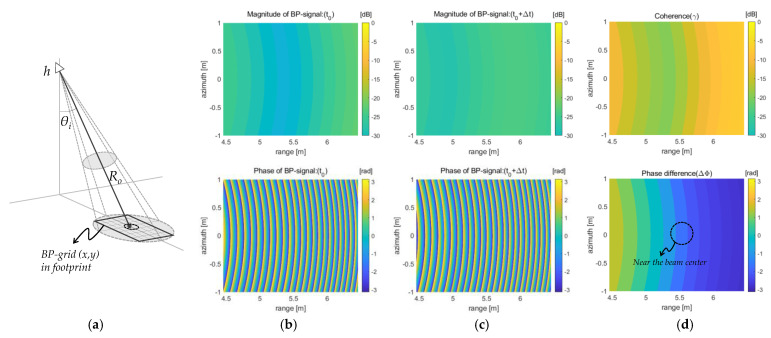
Process of extracting the interferometric phase data using the back-projected signals onto the reference grid: (**a**) the radar geometry for the back-projection; (**b**,**c**) the back-projected signals’ magnitude and phase terms at the initial time step (*t*_0_) and the next time step (*t*_0_ + Δ*t*); (**d**) the Pearson’s product–moment correlation coefficients (or coherence map) and the self-interferometric phase.

**Figure 6 sensors-23-03049-f006:**
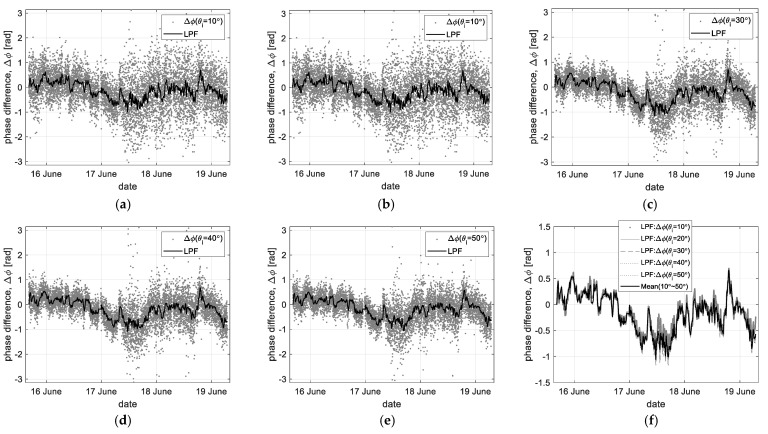
Time-series self-interferometric phase data (ΔΦ) and its low-pass filtered data (‘LPF’): (**a**–**e**) the original & filtered data at each incident angle, and (**f**) the comparison of all ‘LPF’ data.

**Figure 7 sensors-23-03049-f007:**
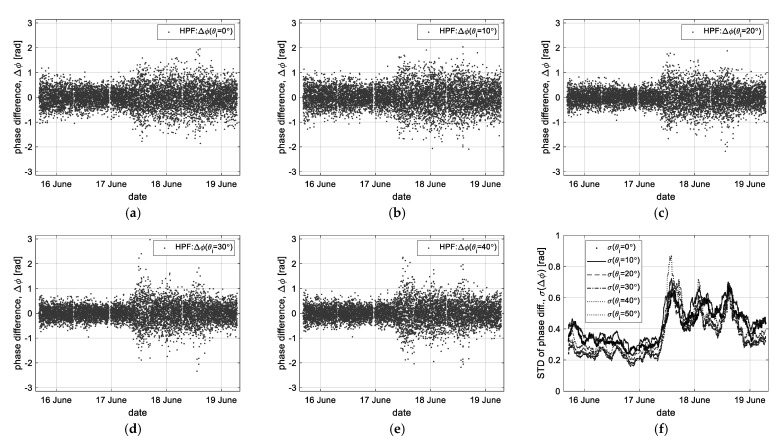
High-pass filtered data (‘HPF’) only of the time-series interferometric phase data (ΔΦ): (**a**–**e**) the filtered data at each incident angle, and (**f**) the standard deviation of HPF data (*σ*(ΔΦ)).

**Figure 8 sensors-23-03049-f008:**
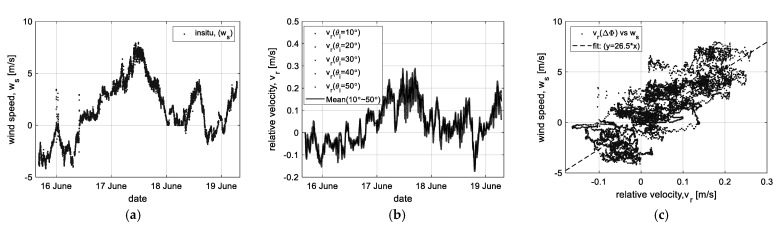
Correlation analysis of the relative velocity (*v_r_*) versus wind speed (*w_s_*): (**a**) the in situ data (*w_s_*), (**b**) the relative velocity (*v_r_*), and (**c**) the result of correlation analysis.

**Figure 9 sensors-23-03049-f009:**
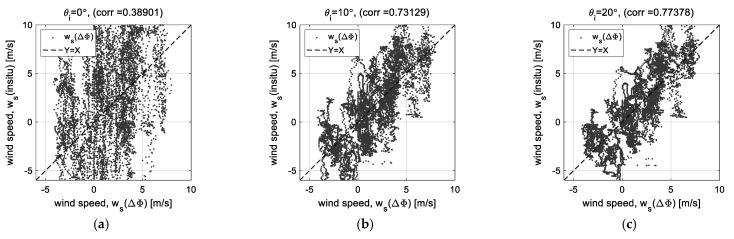
Results of the linear regression analysis of self-interferometric phase (ΔΦ) to wind speed: (**a**–**f**) correlation diagrams at each incident angle.

**Figure 10 sensors-23-03049-f010:**
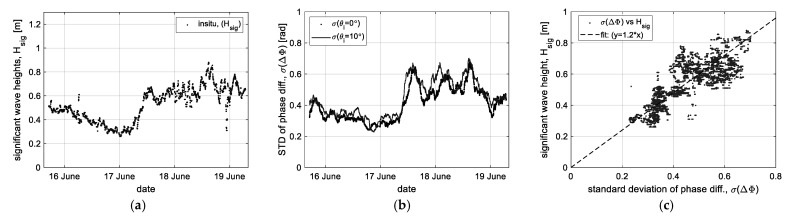
Correlation analysis of the ‘HPF’ data (ΔΦ) versus the significant wave height (*H_sig_*): (**a**) the in situ data of significant wave height, (**b**) the data (*σ*(ΔΦ)) reprocessed by statistical analysis, and (**c**) correlation between the relative velocity and significant wave height.

**Figure 11 sensors-23-03049-f011:**
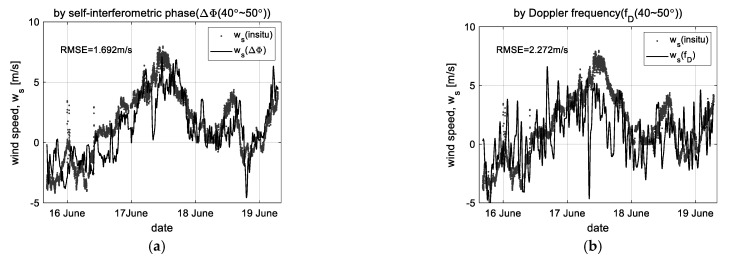
Observation results for wind speed(*w_s_*): (**a**,**b**) the wind speed retrieved by the proposed- and existing method at the unstable region (40°~50°), (**c**) the observation result of wind speed by the proposed method, and (**d**) the comparison of correlation coefficients (ΔΦ vs. *f_D_*).

**Figure 12 sensors-23-03049-f012:**
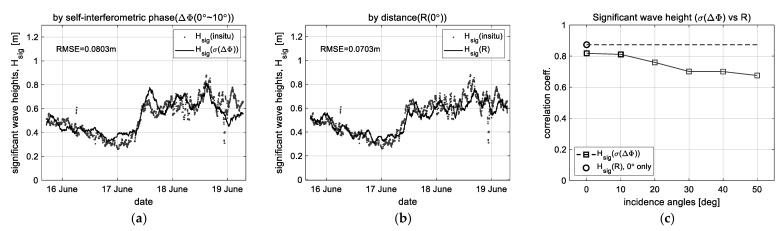
Observation results for significant wave height (*H_sig_*): (**a**,**b**) the significant wave heights retrieved by the proposed and existing method, and (**c**) the comparison of correlation coefficients (*σ*(ΔΦ) vs*. R*).

**Table 1 sensors-23-03049-t001:** Specification of the self-manufactured FMCW scatterometer system (* time interval for analyzing the self-interferometric phase) [11].

System Parameters	Specifications	Notes
Operating frequency	*f*_0_ = 9.65 GHz	X-band
Bandwidth	*BW* = 498 MHz	Chirp rate, *K_r_* ≈ 498 × 10^9^ Hz/s
Tx power	*P_max_* = 3 Watts	-
Range resolution	Δ*x* = 0.3 m	Δx = c/2 BW, (*c* light speed)
Fast-time sampling	*f_ADC_* = 1.2 MHz	1252 samples (=1 array)
Slow-time sampling	*f_PRF_* = 100 Hz	* Δ*t* = 10 ms, 100 arrays
Incident angle	*θ_i_* = 0°~50°, *ϕ_az_* = 0°, 90°	Look dir. East & South
Tx/Rx antenna	*θ* = 12°, *ϕ* = 10°	Half Power BeamWidth
Polarization	Vertical	-

**Table 2 sensors-23-03049-t002:** Comparison of the observational performance for wind speed (*w_s_*) and significant wave height (*H_sig_*): (^(2)^ before and ^(1)^ after improving the performance at the high incident angles).

Radar Parameters	Obs. Parameters	Incident Angle	RMS Error	Corr. Coeff.	Notes
Self-interferometric phase (ΔΦ)	Wind speed (*w_s_*)	10°~50°	1.731 m/s	0.776	(proposed)
		^(1)^ (40°~50°)	(1.692 m/s)	(0.790)	
	Sig. wave height (*H_sig_*)	0° ~10°	8.03 cm	0.814	
Doppler frequency (*f_D_*)	Wind speed (*w_s_*)	10°~30°	1.782 m/s	0.791	(existing)
		^(2)^ (40°~50°)	(2.272 m/s)	(0.453)	
Distance (*R*)	Sig. wave height (*H_sig_*)	0°	7.02 cm	0.873	

## Data Availability

Data is contained within the article.

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
