# Peer review of "Back-Projected Signal-Based Self-Interferometric Phase Analysis Technique for Sea Surface Observation Using a Single Scatterometer System"

_sensors, 2023, doi:10.3390/s23063049_

Round 1
Reviewer 1 Report (New Reviewer)
The instrumentation and signal processing are not well described.
In the title and the text, there is mention FMCW system, but more information is missing. In equation (1) the impact of the FMCW modulation is neglected, and further processing is derived for the CW signal. It seems that FMCW modulation was used for receiver design reasons only to avoid signal mixing to the baseband.
The authors wrote that the signal is not processed continuously but for sections (time intervals). Does those time intervals relate to the chirp rate?
What does mean bandwidth in Table 1. Is it Chirp signal bandwidth, of IF bandwidth, or...?
In table 1 and Figure 4 authors mention range resolution. The range resolution probably depends on the correlation property of the FMCW signal and signal processing; details are missing.
The scale of magnitudes in figure 5 is too large.
The comparison of the original amplitude-based and proposed phase-based signal processing in sense of sensitivity or processing gain should be added as well as the power budget of the scatterometer as function of antenna high and incidence angle.
The authors should extend the paper of missing information. A new revision is needed.
Author Response
Authors are deeply thanks for the thoughtful review.
Major revision was conducted in accordance with the reviewer’s comments, and the revision contents are as follows.
- We agree with the review comment the ‘FMCW’ is not had a special meaning in this manuscript because this technique is established regardless of the FMCW radar or even the pulse radar. So, ‘FMCW’ was removed from the title and abstract.
-->pp.1, title & abstract
- This scatterometer system consists of the FMCW-radar transceiver, which has a Pulse Repetition Frequency of 100Hz. The ‘time interval’ means the sampling interval(=1/PRF) of the consecutive Rx signals. And also, the ‘bandwidth’ means the chirp signal bandwidth, where the chirp rate(Kr) is about 498e9Hz/s. (where the sweep time for chirping is about 1ms)
-->pp.3, table 1
- In a conventional interferometric system, the range resolution and the baseline length affect the interferometry analysis range and resolution, respectively. For this research, the range resolution(δR) relates to the physical pixel size for the interferometric phase analysis, and the time interval(Δt) is corresponding to the baseline. In other words, the range resolution is related to the number of valid scatterers within a footprint and the ambiguity-free range of the target’s physical change. And the time interval also affects the analytic resolution of the interferometric phase corresponding to the sea surface movements. For example, for the same movement, applying the shorter time interval can precisely analyze a tiny change on the sea surface, while applying the longer time interval can roughly analyze a relatively large-scale change. This explanation was described in section 3.
--> pp.7, section 3
- This is just drawn to explain the phase difference component extraction process using the back-projected signals and is shown without the calibration process because the phase component is of primary interest. The magnitude of the back-projected signal can replace with the normalized value. (where the back-projected signal is not necessary to be an absolute value) However, the calibration process is surely needed to get the backscattering coefficient(σº) as in the previous study[11].
-->pp.9, figure 5
Reviewer’s comments helped us very much for our research. Thanks a lot.
Reviewer 2 Report (New Reviewer)
1. In the work under review, a method of self-interferometric phase analysis was developed for observing the sea surface and determining such quantities as wind speed and wave height. When constructing the method, the back projection algorithm, and a series of successive signals from one system of reflectometer without any auxiliary system or channel were used.
2. The advantage of the method proposed by the authors is its efficiency at relatively large angles of deviation of the scanning system from the vertical.
3. The proposed method was tested from a stationary platform. It is not clear if the method can be applied from a moving platform? If such a generalization were made, the relevance of the work would increase significantly.
Remarks:
4. Unfortunately, the introduction does not describe the content of the work by sections, as the authors usually do. This would make the work easier to understand.
5. What was meant by the term "exiting method" (line 128)? Could it be "an existing method"?
6 It was necessary to describe in more detail the designations in Fig. 4. This is especially true for Fig. 4 с.
7. Figures 6 f and 7 f are too small, the curves corresponding to different angles are hard to distinguish.
8. The paper does not indicate the frequency at which observations were made. How were filtering frequencies chosen for LPF and HPF?
9. It is not clear from the work whether the results of the work confirm the data measured in other months and days? Have such observations been made?
10. It is not clear what the authors meant by capitalizing the article "The" in the middle of the phrase (line 147).
11. In Section 5, the subsections are labeled 6.2 and 6.2. Is this a mistake?
Author Response
Authors are deeply thanks for the thoughtful review.
Major revision was conducted in accordance with the reviewer’s comments, and the revision contents are as follows.
1~3. The back-projection algorithm has merit for the platform motion errors if it is applied with the inertial navigation system. So, we are also studying now how to apply the proposed method to the airborne system. In particular, we are constantly researching a method for precisely compensating the platform movement in the proposed technique for ocean wind retrieval. Thanks for your thoughtful comment.
- The content of the work by sections was additionally described in the introduction.
--> pp., introduction
- The typo error was corrected.
-->pp.5, figure 2 (line 127)
- The additional explanations were added in section 3. --> pp.5, section3
- Figure 6f shows that the low pass filtered data has a quite similar trend reflecting a horizontal movement of the sea surface regardless of the incident angles. And figure 7f also shows that the data applying the high pass filtering & weight factor(e.g., 1/cosθi) has a similar trend corresponding to a vertical movement, too.
--> pp.9, figure 6(f); pp.10, figure 7(f)
- the explanations about setup parameters of the digital filter were added in section 4.2
--> pp.10, section 4.2 (line 295~301)
- We conducted field campaigns for several years and collected the raw data measuring the sea surface. For this study, the data measured in 2017 was mainly used to verify the proposed analysis technique because the capillary wave was dominant. Whereas, other data sets measured under rough sea state, more than WMO sea state code 4, were not suitable for this. (please refer to the previous study [11]) The following figure shows the analysis result and in-situ data(e.g, significant wave height, wind direction and speed). This result was analyzed by the same proposed method or setup using the data collected under the ‘moderate’ sea state. It also agrees well with in-situ data relatively.
10~11. The typo errors were corrected.
--> pp.5, line 145
-->pp.13, line 391; pp.14, line 416
Reviewer’s comments helped us very much with our research. Thanks a lot.
Round 2
Reviewer 1 Report (New Reviewer)
The reviewer's comments have been considered in the revised manuscript. The manuscript is ready for publication.
This manuscript is a resubmission of an earlier submission. The following is a list of the peer review reports and author responses from that submission.
Round 1
Reviewer 1 Report
1. The manuscript is concerned with back-projected signal based time-series interferometric phase analysis technique for ocean observation using FMCW-scatterometer system. It is relevant and within the scope of the journal.
2. However, the manuscript, in its present form, contains several weaknesses. Adequate revisions to the following points should be undertaken in order to justify recommendation for publication.
3. Full names should be shown for all abbreviations in their first occurrence in texts. For example, WMO in p.2, RMS in p.11, etc.
4. For readers to quickly catch the contribution in this work, it would be better to highlight major difficulties and challenges, and your original achievements to overcome them, in a clearer way in abstract and introduction.
5. It is shown in the reference list that the authors have a pertinent publication in this field. This raises some concerns regarding the potential overlap with their previous works. The authors should explicitly state the novel contribution of this work, the similarities, and the differences of this work with their previous publications.
6. p.1 - an FMCW signal-based time-series interferometric phase model is adopted for ocean observation. What are the other feasible alternatives? What are the advantages of adopting this model over others in this case? How will this affect the results? The authors should provide more details on this.
7. p.2 - the Ieodo Ocean Research Station is adopted as the case study. What are other feasible alternatives? What are the advantages of adopting this case study over others in this case? How will this affect the results? The authors should provide more details on this.
8. p.10 - the linear regression analysis is adopted in the analysis. What are the other feasible alternatives? What are the advantages of adopting this modeling technique over others in this case? How will this affect the results? The authors should provide more details on this.
9. p.10 - the standard deviation is adopted as a statistics value. What are other feasible alternatives? What are the advantages of adopting this indicator over others in this case? How will this affect the results? The authors should provide more details on this.
10. p.11 - RMS error is adopted to evaluate the performance of the model. What are the other feasible alternatives? What are the advantages of adopting this evaluation metric over others in this case? How will this affect the results? More details should be furnished.
11. p.12 - Doppler frequency and distance as shown in Table 2 are adopted as benchmarks for comparison. What are the other feasible alternatives? What are the advantages of adopting these methods over others in this case? How will this affect the results? More details should be furnished.
12. Some key parameters are not mentioned. The rationale on the choice of the set of parameters should be explained with more details. Have the authors experimented with other sets of values? What are the sensitivities of these parameters on the results?
13. The discussion section in the present form is relatively weak and should be strengthened with more details and justifications.
14. Some assumptions are stated in various sections. More justifications should be provided on these assumptions. Evaluation on how they will affect the results should be made.
15. Moreover, the manuscript could be substantially improved by relying and citing more on recent literature about real-life applications of time-series modeling techniques such as the following. Discussions about result comparison and/or incorporation of those concepts in your works are encouraged:
● Alassafi, M.O., et al., “Time series predicting of COVID-19 based on deep learning,” Neurocomputing 468: 335-344 2022.
● Miky, Y., et al., “A Recurrent-Cascade-Neural network-nonlinear autoregressive networks with exogenous inputs (NARX) approach for long-term time-series prediction of wave height based on wave characteristics measurements,” Ocean Engineering 240: 109958 2021.
● , et al., “Time series-based groundwater level forecasting using gated recurrent unit deep neural networks,” Engineering Applications of Computational Fluid Mechanics 16 (1): 1655-1672 2022.
16. Some inconsistencies and minor errors that needed attention are:
● Replace “…And also, confirmed that this…” with “…And also, we confirmed that this…” in line 19 of p.1
● Replace “…Station (refer to figure 1…” with “…Station (refer to Figure 1…” in line 68 of p.2
● Replace “…calculated by equation (13)…” with “…calculated by Equation (13)…” in line 305 of p.10
● and more…
17. In the conclusion section, the limitations of this study, suggested improvements of this work and future directions should be highlighted.
Author Response
Authors are deeply thanks for the thoughtful review.
First revision was conducted in accordance with the review comments below, and the revision contents are as follows attached file.

Reviewer 2 Report
This work reports an interferometric phase detection method to determine the wind velocity and wave height for oceanographic analysis. The results reported here are of wide interest to the research community. Although minor grammatical corrections are required, but overall the article is well written. The results are appropriately presented to justify the claim.
In my opinion, the article must be published with minor corrections as described below.
I suggest to recheck and correct all the units along y-axis in Figure-3(b-d). The colorbar shown in Fig.3(b-d) should have the same limits for both the upper and lower components in each part. Also mention the units of data shown in each colorbar in the caption.
Author Response

(The authors gave the same response as above.)

Reviewer 3 Report
The time-series interferometric phase analysis technique for ocean observation using a single FMCW-scatterometer system is presented in this paper. The whole process for extracting the interferometric phase components from the scatterometer’s raw data is theoretically verified by the back-projected signal model. The paper is well written. I have some minor questions as follows:
1) Each observational result has RMS errors of 1.69m/s and 11.72 cm for wind speed and significant wave height, respectively. If the author can give some explanations affecting the accuracy error, it may be more complete.
2) What the depth of the ocean in the measured area? The depth of ocean may affect the accuracy of the observational result, as discussed in some articles.
3) There are some errors in equation (3)
4) In line 125, there are some grammatical mistakes in the sentence “ The distance between the target and the antenna can calculate using the defined points in Equation (3)”.
5) The baseline plays an important role in the processing. How to choose the baseline? In other words, how to choose △t?
Author Response

(The authors gave the same response as above.)

Round 2
Reviewer 1 Report
Enough chances have been given to the authors to revise the manuscript but no improvement has been made. The most significant comments in the previous reviews (including novelty, major difficulties and challenges, their original achievements to overcome them, overlapping with their previous works, etc.) have not been demonstrated satisfactorily. The overlapping with their very recent work [8] cannot be totally neglected.
Author Response
Dear Reviewer
Authors are deeply thanks again for the thoughtful review.
First of all, We regret that there were some insufficient parts in the first revision. So, intensively revised those parts, such as the Title, Abstract, Introduction, and Figures 1 & 2 in Sections 2 & 3.
Of them, Introduction was particularly re-written to clarify the purpose or advantages of this study.
Figure 1(c) was newly added to complement the insufficient explanations about the time-series interferometric phase measurable with a single channel or system. And also, Figure 2 was revised and moved to section 3, and explanations of the field campaign conducted at the Ieodo Ocean Research Station were added.
In the Title, ‘Ocean observation’ was replaced with ‘sea surface observation’ to express the observation object specifically.
In addition, the existing references that became unnecessary due to the revision of the Introduction were removed and new references were added, and all references were rearranged.
As minor revisions, sentences with unclear explanations or redundancy have been deleted or modified. Lower subscripts (t0) or (t0+Δt) in Equations (3)~(11) were also simplified to (1) or (2). And the captions of Figures 4~6 were partially corrected.